# Application of Graphene in Acoustoelectronics

**DOI:** 10.3390/nano14211720

**Published:** 2024-10-28

**Authors:** Dmitry Roshchupkin, Oleg Kononenko, Viktor Matveev, Kirill Pundikov, Evgenii Emelin

**Affiliations:** Institute of Microelectronics Technology and High Purity Materials Russian Academy of Sciences, 142432 Chernogolovka, Russia; oleg@iptm.ru (O.K.); matveev@iptm.ru (V.M.); pundikov@iptm.ru (K.P.); eemelin@iptm.ru (E.E.)

**Keywords:** graphene, multilayer graphene, surface acoustic wave, interdigital transducer, scanning electron microscopy

## Abstract

An interdigital transducer structure was fabricated from multilayer graphene on the surface of the YZ-cut of a LiNbO_3_ ferroelectric crystal. The multilayer graphene was prepared by CVD method and transferred onto the surface of the LiNbO_3_ substrate. The properties of the multilayer graphene film were studied by Raman spectroscopy. A multilayer graphene (MLG) interdigital transducer (IDT) structure for surface acoustic wave (SAW) excitation with a wavelength of Λ=60 μm was fabricated on the surface of the LiNbO_3_ crystal using electron beam lithography (EBL) and plasma chemical etching. The amplitude–frequency response of the SAW delay time line was measured. The process of SAW excitation by graphene IDT was visualized by scanning electron microscopy. It was demonstrated that the increase in the SAW velocity using graphene was related to the minimization of the IDT mass.

## 1. Introduction

The future prospects of modern micro- and nanoelectronics are related to the use of low-dimensional materials. The use of graphene is a good example. Among various carbon structures (carbon nanotubes, fullerenes, graphite, diamante, diamond, etc.), graphene is a monoatomic two-dimensional $sp^{2}$ hybridized carbon layer with unique physical properties [1,2,3,4]. Interest in graphene developed immediately after its discovery in 2004 [5]. Various methods can be used to produce graphene, one of which is graphite splitting [5]. In this method, the area of graphene is a few tens of μm^2^. For practical applications in modern micro- and nanoelectronics technologies, it is necessary to have graphene films with large areas (tens of cm^2^).

There are several methods that allow the formation of large-area graphene films. The most used method is CVD (chemical vapor deposition). The process of graphene growth is carried out using a metal catalyst (Ni, Fe, Cu films) on SiO_2_/Si substrate. It has been shown that the quality of the graphene film is determined by the quality of the metallic film on the substrate surface [6,7,8,9,10]. The growth of graphene occurs in several stages. The first step involves the decomposition of the carbon-containing precursor. The second step involves diffusion of carbon into the metallic film. The third and final step involves segregation and deposition of carbon on the surface of the metallic film. The presence of grain boundaries and dislocations in a metallic film leads to the accumulation of carbon on them and, consequently, to the formation of multilayer graphene [6,7,8,9]. The graphene film is then transferred to other substrates. For this purpose, a polymethyl methacrylate (PMMA) layer is deposited on the substrate surface with a metallic film and graphene, which acts as a supporting backing film. The graphene along with the PMMA backing film is separated from the substrate by liquid etching of the metallic film in 1% aqueous hydrochloric acid solution, and then transferred onto the substrate for further application. PMMA is removed from the graphene surface by dissolution in acetone.

It is also possible to directly synthesize graphene by CVD or molecular beam epitaxy (MBE) on the substrates. In [10], the possibility of direct CVD synthesis of graphene film on the surface of the X-cut of a La_3_Ga_5.5_Ta_0.5_O_14_ (LGT) piezoelectric crystal was demonstrated. The LGT crystal belongs to the family of lanthangallium silicate group crystals and possesses the spatial symmetry group 32 [11]. The possibility of the direct synthesis of graphene on the surface of piezoelectric crystals of the langasite family is associated with a good matching of the crystal unit cell parameters of the crystal and graphene [10], which opens new possibilities for the use of graphene in opto- and acoustoelectronics [12,13] as transparent electrodes.

Currently, graphene is already used in microelectronics [14]. In [15], the possibility of using graphene in the creation of ultra-high frequency (UHF) transistors was demonstrated. In addition, it is worth noting the application of graphene in acoustoelectronics for acoustically stimulated charge transport [16,17,18,19,20,21,22,23,24,25,26,27,28,29,30,31,32]. Here, graphene is used as a medium for charge transport under the influence of a traveling surface acoustic wave (SAW). Accordingly, electrons and holes are redistributed between the SAW minima and maxima and move across the substrate surface with the SAW velocity.

The fabrication of interdigital transducers (IDTs) made of graphene for SAW excitation on the surface of piezoelectric crystals is also of interest [19,33,34]. Also, the creation of graphene structures is important for the development of X-ray acousto-optics, where it is possible to obtain the ultra-short pulses of X-ray radiation on laboratory and synchrotron radiation sources in the process of X-ray diffraction on the piezoelectric crystal surface by applying pulses of external electric field to the IDT structure on the crystal surface [35,36].

However, the use of graphene for the fabrication of acoustoelectronic devices is still difficult due to the high electrical resistance of graphene films compared to Al used in acoustoelectronics, and problems in the fabrication of IDT structures. It is also necessary to take into account the differences in mass and mechanical load on the surface of piezoelectric crystal caused by IDTs of Al and graphene.

This paper considers the technology of the fabrication of IDT structures from graphene. The acoustic characteristics of IDTs made of Al and graphene are compared. Also, the process of SAW excitation using Al and graphene IDTs is investigated by scanning electron microscopy (SEM).

## 2. Fabrication of Graphene IDTs for the SAW Delay Time Line

To investigate the possibility of SAW excitation by graphene IDTs, the surface acoustic wave delay time lines were fabricated on the surface of the YZ-cut of a LiNbO_3_ crystal. For comparison studies, the IDT structures were fabricated from multilayer graphene and Al.

Multilayer graphene films were synthesized by CVD at low pressure with a single acetylene injection into a vacuum chamber [10]. Films of pure Fe with a thickness of 0.3 µm sputtered on the oxidized silicon wafers by electron beam deposition were used as a catalyst. After synthesis of MLG, a 0.5 µm thick PMMA layer was deposited on the surface of the iron films with the grown multilayer graphene, which served as a support film for the graphene. The samples with the multilayer graphene film were then immersed in a one percent water solution of hydrochloric acid to dissolve the iron film. During the etching process, the graphene, along with the supporting PMMA film, was separated from the substrate. After complete dissolution of the iron, the graphene with PMMA was washed in deionized water and transferred to the substrate surface of the YZ-cut of a LiNbO_3_ crystal.

Raman spectroscopy was used to characterize the graphene films. The spectra were measured using a SENTERRA Bruker Raman microscope with a laser wavelength of 532 nm. The diameter of the laser beam was ~1.5 µm. Figure 1 shows a typical Raman spectrum obtained for a graphene film on the LiNbO_3_ crystal surface (blue line). The spectrum shows clear G and 2D peaks indicating the presence of graphene. A weak D peak indicates low defect content. The ratio of the intensities of the 2D and G peaks equal to 0.5 indicates the multilayer graphene film (MLG).

Figure 2 shows the schemes of the fabrication technology of Al and MLG structures on the surface of a LiNbO_3_ crystal. Figure 2a shows the scheme of the fabrication of an IDT structure made of Al by electron beam lithography (EBL). Firstly, PMMA 950 resist was deposited on the surface of the YZ-cut of a LiNbO_3_ crystal, in which the IDT structure was formed. After the EBL and resist development, a 100 nm thick Al layer was sputtered on the crystal surface by magnetron sputtering. In the final step, a “lift-off” operation in dimethylformamide was carried out, after which the *Al*-IDT structure remained on the crystal surface.

Figure 2b shows the step-by-step fabrication technique for IDTs made of multilayer graphene. In the first step, an MLG film was transferred onto the surface of the YZ-cut of a LiNbO_3_ crystal. Then, a layer of PMMA 950 resist was deposited on the MLG surface by centrifugation. Next, the topology of the future IDT structure was formed in the resist layer by EBL. Then, a 40 nm thick MgO film was deposited by magnetron sputtering. In the next step, a “lift-off” operation was performed, and a protective MgO structure in the form of the IDT was left on the MLG surface. The MLG film was then etched in oxygen plasma through the MgO mask down to the substrate surface of the YZ-cut of a LiNbO_3_ crystal. And in the final step, the protective MgO film was removed in a 1% water solution of HCl, leaving the IDT structure of MLG on the surface of the substrate. Multilayer graphene was controlled at all stages of fabrication by Raman spectroscopy. The spectra at all stages of fabrication were in good agreement with each other (Figure 1), which indicated that there was no influence exerted by technological operations on the properties of graphene. In Figure 1, the red line corresponds to the Raman spectrum of *MLT*-IDT. As can be seen from the figure, the Raman spectrum of *MLT*-IDT corresponds to the spectrum of the original multilayer graphene on the LiNbO_3_ crystal surface. Thus, the technological operations performed for IDT fabrication do not affect the properties of MLG.

Figure 3 shows optical photographs of IDTs fabricated on the surface of the YZ-cut of a LiNbO_3_ crystal. Figure 3a demonstrates an IDT fabricated from Al. The number of electrode pairs is 25. The period of the Al electrodes is 30 μm with a width of one electrode of 15 μm. Specifically, this IDT is designed for the excitation of a SAW with a wavelength of Λ=60 μm. The IDT structure made of Al has a perfect appearance. Figure 3b,c show the microphotographs of the IDT fabricated from MLG. It is evident that graphene is not yet a good technological material like Al. At low magnification (Figure 3b), the structure of *MLG*-IDT follows that of *Al*-IDT. At higher magnification (Figure 3c), it can be observed that the MLG film on the IDT electrodes consists of small flakes. However, in general, it can be stated that it was possible to form an IDT structure from MLG. Additionally, to drive the high-frequency electrical signal to the *MLG*-IDT, square Au electrodes were sputtered onto the contact pads.

The fabricated *MLG*-IDT structure allowed us to measure the film thickness of multilayer graphene. Figure 4 shows the *MLG*-IDT electrode profile measured by atomic force microscopy (AFM). The figure shows that the height of the *MLG* electrode is approximately 20 nm, which corresponds to a thickness of ~60 layers of graphene.

## 3. Investigation of SAW Excitation and Propagation

The process of SAW excitation and propagation on the surface of the YZ-cut of a LiNbO_3_ crystal using standard *Al*-IDT and *MLG*-IDT was investigated by measuring the amplitude–frequency responses and visualization of acoustic wave fields on a crystal surface by scanning electron microscopy (SEM). A comparison of the results for IDTs fabricated from Al and MLG highlights the peculiarities of SAW excitation and propagation with IDT fabricated from multilayer graphene.

Figure 5 shows the amplitude–frequency responses S11 for both fabricated IDTs measured with a spectrum analyzer. From Figure 5 it can be seen that the resonance excitation frequency of the SAW with a wavelength of Λ=60 μm using *Al*-IDT is $f_{0}=57.0$ MHz. The SAW velocity in this case is $V=\Lambda \cdot f_{0}=60\cdot 57.0=3420$ m/s.

In the case of *MLG*-IDT, the resonance excitation frequency of the SAW is $f_{0}=58.8$ MHz, which corresponds to the SAW propagation velocity of $V=\Lambda \cdot f_{0}=60\cdot 58.8=3528$ m/s. The difference in the SAW excitation frequencies is due to the difference in the masses of the Al and MLG IDTs and, respectively, the different mechanical loading on the surface of the ferroelectric LiNbO_3_ crystal. It is well known that SAW velocities differ on free and metallized surfaces; this is used to determine the electromechanical coupling coefficients. It was also shown in [37] that changing of the metallization coefficient of the IDT leads to a change in frequency as the mass of the IDT is changed. Accordingly, as the metallization coefficient increases, the IDT mass increases and the loading on the crystal surface increases; this in turn leads to a decrease in the SAW excitation frequency and velocity. In our case, geometrically identical IDTs made of Al and graphene have different masses due to the materials used; this results in different SAW velocities. The mass of *Al*-IDTs exceeds the mass of *MLG*-IDTs, which leads to an increase in the SAW velocity for the MLG.

Also, Figure 5 shows that the acoustic loss for *MLG*-IDT is almost a hundred times higher than that for *Al*-IDT (factor ×100 in Figure 5). This is because the graphene has a high electrical resistivity [19,33,34,38], which is significantly higher than the electrical resistivity of Al film, which has a thickness of ~100 nm and is commonly used in the process of IDT fabrication. While the electrical resistivity of Al film is ~2.7 × 10^−8^ Ω·m, the electrical resistivity of MLG is only ~4.2 × 10^−6^ Ω·m, which leads to large losses in the process of SAW excitation and a decrease in the quality factor of the acoustoelectronic device.

Two approaches can be used to improve the electrical and, respectively, the acoustic characteristics of *MLG*-IDT. The first approach is related to the doping of graphene, which leads to an increase in the conductivity and a corresponding decrease in the resistivity of graphene. However, the doped state of graphene is metastable and cannot exist for a long time. Over time, graphene reverts back to its original state. Graphene encapsulation also fails to significantly improve the electrical properties of *MLG*-IDT. There is a second approach to improve the electrical and acoustic characteristics of *MLG*-IDT used in [19,33], where the length of the *MLG*-IDT pins is reduced in order to reduce the IDT resistivity. In this case, the losses caused by high impedance are reduced, but there are large losses caused by the diffraction of the surface acoustic wave on the *ML*G-IDT aperture (the analog of light diffraction on a slit). In such cases, the acoustic wave will be divergent at the IDT output, resulting in high acoustic losses. Therefore, in the future, to create a graphene IDT, it will be necessary to solve a serious problem regarding the optimization of IDT parameters (aperture of IDT, number of electrode pairs, and resistivity of graphene).

Moreover, the process of SAW excitation and propagation by graphene IDT has been investigated by SEM, which allows a real-time visualization of the SAW on the surface of piezo- and ferroelectric crystals [39,40,41,42,43,44]. For visualization of the SAW, it is necessary to use a low electron accelerating voltage U=1 kV, because at higher accelerating voltages the dielectric substrate surface is negatively charged, which leads to image distortion. The method of the SAW visualization is based on the high-frequency modulation of the low-energy secondary electrons emitted from the crystal surface. When a SAW propagates in the piezoelectric crystals, the maxima and minima of the wave have the opposite potentials. Consequently, the minima and maxima of the SAW have different emission coefficient of low-energy secondary electrons due to the different signs of the potential. While a positive potential leads to a decrease in the emission coefficient, a negative potential conversely leads to an increase in the emission of low-energy secondary electrons from the crystal surface. Thus, a contrast is formed between the minima and maxima of the SAW. Since the SAW is a traveling wave, the formation of a stationary SAW image is due to the effect of autostroboscopy. The IDT excites both the SAW in the crystal and the electric field above the crystal surface. The SAW and the electric field are mutually coherent because they are emitted by the same source (IDT) and at the same frequency. These two wave processes differ only in their velocities. If the propagation velocity of the SAW in the LiNbO_3_ crystal is about 3500 m/s, the propagation velocity of the electric field corresponds to the velocity of light (c≈300,000,000 m/s). And accordingly, the electric field wavelength is almost five orders of magnitude higher than the SAW wavelength. Respectively, when the crystal surface is covered by a positive half-wave of the electric field, the low-energy secondary electrons reach the secondary electron detector, while in the case of a negative half-wave, the secondary electrons do not reach the secondary electron detector. Thus, a stationary image of the traveling SAW on the crystal surface is formed in the SEM in the real-time mode. This method is the optimal method for studying the excitation and propagation of bulk, surface and pseudo-surface, traveling, and standing acoustic waves in piezo- and ferroelectric crystals.

Figure 6 shows the SEM micrographs of the SAW excited with *Al*-IDT and *MLG*-IDT. The microphotograph in Figure 6a corresponds to a SAW with a wavelength of Λ=60 μm excited at the resonance frequency of $f_{0}=57.0$ MHz using *Al*-IDT. The resonance excitation frequency of the SAW corresponds to the maximum contrast of the SAW image. Figure 6b shows the SEM micrograph of the SAW image excited at the resonant frequency of $f_{0`}=58.8$ MHz using a graphene interdigital transducer. In this case, the wavelength of the SAW is also Λ=60 μm. Thus, the use of the SEM allows the visualization of the excitation of the SAW by *MLG*-IDT. It is also demonstrated that the resonance excitation frequency of the SAW using *MLG*-IDT is higher than that of the SAW using *Al*-IDT; this is due to the lower mass of *MLG*-IDT and, consequently, the lower mechanical loading on the crystal surface.

## 4. Conclusions

The process of *MLG*-IDT fabrication and SAW excitation was investigated. A comparative analysis of the process of SAW excitation using *MLG*-IDT versus conventional *Al*-IDT was carried out.

The *MLG*-IDT fabrication process consists of further technological operations related to the process of graphene transfer to the piezoelectric substrate and the creation of a protective layer of MgO for the subsequent etching of a graphene film in oxygen plasma. Raman spectroscopy was used during all technological operations of *MLG*-IDT fabrication to control multilayer graphene. It was shown that there was no influence exerted by technological operations on the properties of MLG. To investigate the process of SAW excitation, the amplitude–frequency responses were measured and the SAWs were visualized by SEM.

Comparison of the SAW excitation process demonstrated that the low specific conductivity in graphene leads to high losses in the SAW excitation process and to low quality of the IDT compared to a conventional aluminum IDT. However, it should be noted that the use of graphene can reduce the mass of the IDT and consequently reduce the mechanical loading on the crystal surface, which results in an increase in the SAW velocity on the crystal surface. Thus, the use of *MLG*-IDT increases the frequency of acoustoelectronic devices.

Moreover, it is of interest to use IDTs made of graphene to create sensor devices. For example, [28] demonstrated the possibility of using a graphene SAW resonator as a sensor, including a sensor for TNT detection. In addition, it should be noted that graphene structures can also be used in acousto-optics, where transparent graphene electrodes can be used for SAW excitation.

Graphene IDTs can also be used in adaptive optics, including X-ray optics and acousto-optics, where the application of an external electric field to the graphene structure causes deformation of the crystal lattice due to the inverse piezoelectric effect in the condition of the absence of phase contrast in the process of X-ray diffraction on the graphene film (by analogy with the work [35,36]).

## Figures and Tables

**Figure 1 nanomaterials-14-01720-f001:**
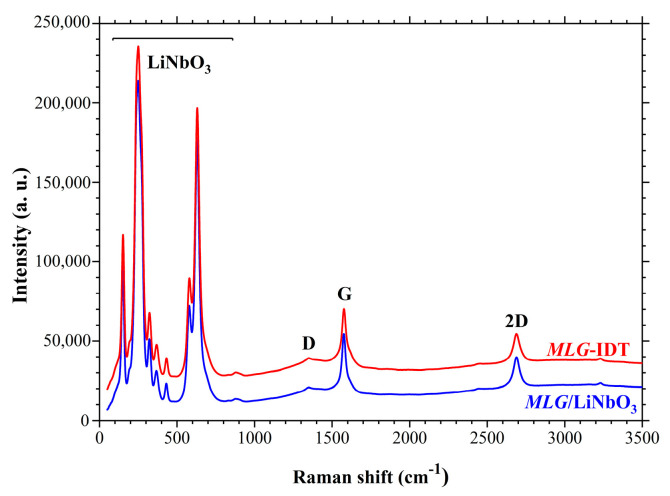
Raman spectra of MLG on the surface of the YZ-cut of a LiNbO_3_ crystal (blue line) and of *MLG*-IDT (red line).

**Figure 2 nanomaterials-14-01720-f002:**
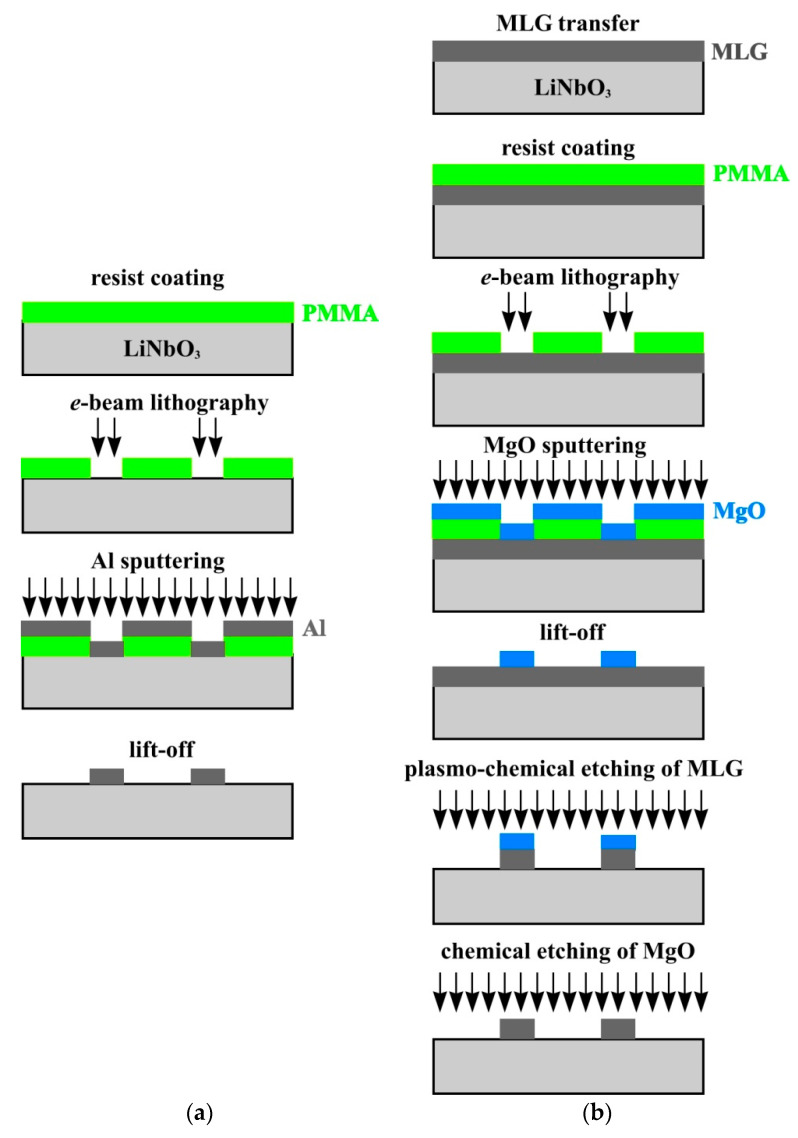
IDT fabrication technology: (**a**) *Al*-IDT; (**b**) *MLG*-IDT.

**Figure 3 nanomaterials-14-01720-f003:**
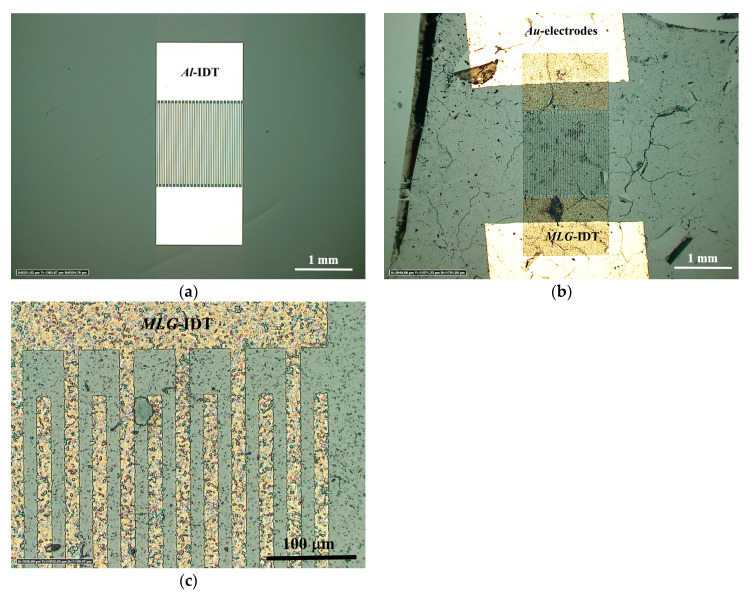
Optical micrographs of IDTs on the surface of the YZ-cut of a LiNbO_3_ crystal: (**a**) *Al*-IDT; (**b**,**c**) *MLG*-IDT. Yellow color corresponds to *MLG*-IDT.

**Figure 4 nanomaterials-14-01720-f004:**
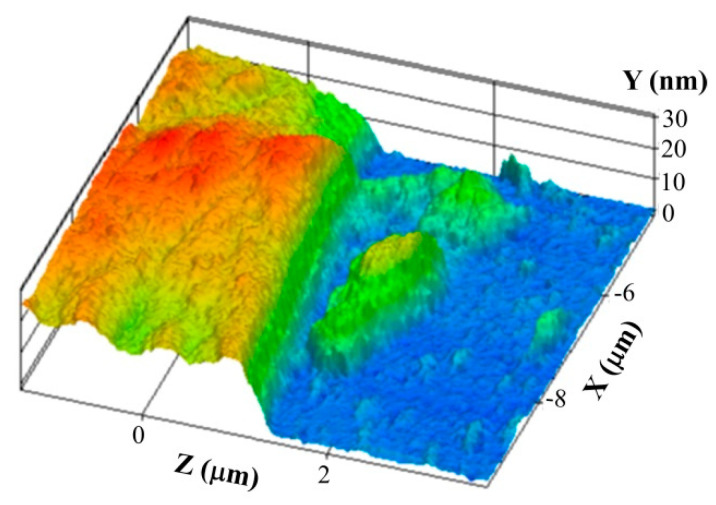
*MLG*-IDT profile measured by the AFM method. The blue color corresponds to the surface of the LiNbO_3_ substrate. Red color corresponds to the surface of *MLG*-IDT electrode.

**Figure 5 nanomaterials-14-01720-f005:**
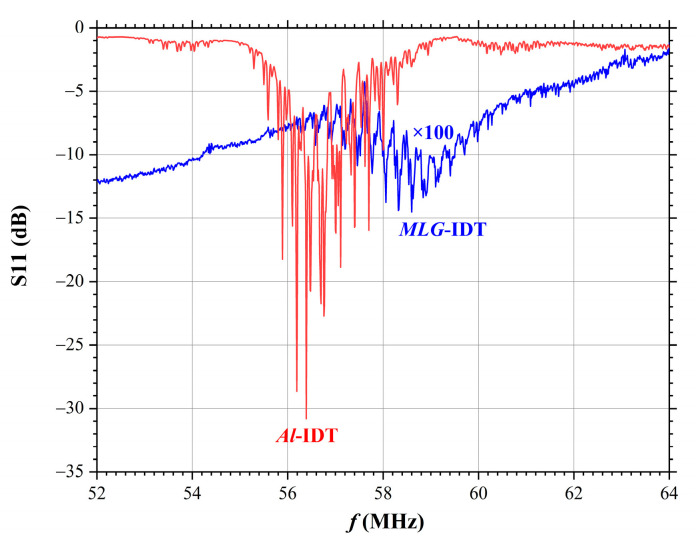
Amplitude–frequency responses of the SAW delay timelines S11, Λ=60 µm: *Al*-IDT (red line); *MLG*-IDT (blue line).

**Figure 6 nanomaterials-14-01720-f006:**
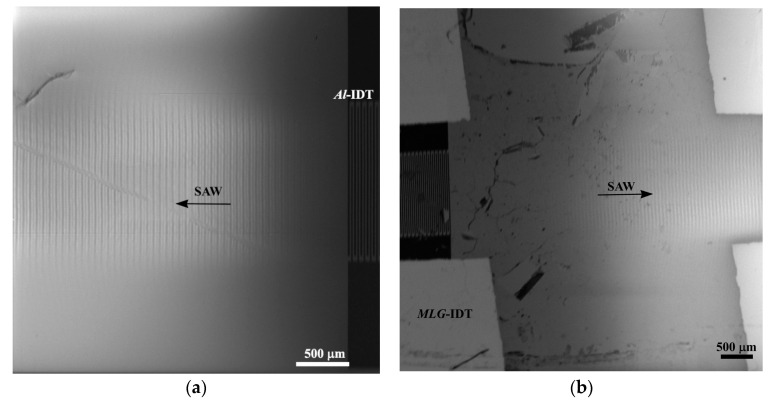
SEM microphotographs of the SAW propagation in the YZ-cut of a LiNbO_3_ crystal, Λ=60 µm: (**a**) *Al*-IDT, $f_{0}=57.0$ MHz; (**b**) *MLG*-IDT, $f_{0}=58.8$ MHz.

## Data Availability

All relevant data presented in the article are stored according to institutional requirements and as such are not available online. However, all data used in this manuscript can be made available upon request to the authors.

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
