# Peer review of "Application of Graphene in Acoustoelectronics"

_nanomaterials, 2024, doi:10.3390/nano14211720_

Round 1
Reviewer 1 Report
Comments and Suggestions for Authors
This manuscript reports on the application of multilayer graphene as interdigital-transducer material in acoustoelectronics. The excitation of surface acoustic waves in such a graphene-based device is compared to that in a conventional Al-based device.
The experiments seem to have been conducted carefully and the interpretation of the obtained data is clear and convincing. In my opinion, the conclusions are adequately supported by the data provided. The work is original and sufficiently well referenced.
I find the present paper suitable for publication in Nanomaterials after a minor revision to correct one inaccuracy pointed out below:
In their comment about the electrical characteristics of the Al film and the multilayer graphene (lines 174 – 177 in the manuscript), the authors use values and units of resistivity rather than conductivity.
Author Response
Dear Reviewer!
Thank you for your comments and helpful remarks. We have modified the text of the article accordingly.
Comment (1): In their comment about the electrical characteristics of the Al film and the multilayer graphene (lines 174 – 177 in the manuscript), the authors use values and units of resistivity rather than conductivity.
Response to comment (1)
We replaced “conductivity” by “resistivity”/
Also Fig. 5 shows that the acoustic loss for MLG-IDT is almost a hundred times higher than that for Al-IDT (factor ×100 in Fig. 5). This is because the graphene has a high electrical resistivity [19, 33-35, 39], which is significantly high than the electrical resistivity of Al film, which has a thickness of ~100 nm and is commonly used in the process of IDT fabrication. While the electrical resistivity of Al film is 2.7×10-8 Ω·m, the electrical resistivity of MLG is only 4.2×10-6 Ω·m, which leads to large losses in the process of the SAW excitation and a decrease in the quality factor of the acoustoelectronic device.

Reviewer 2 Report
Comments and Suggestions for Authors
The work entitled Application of graphene in acoustoelectronics by Roshchupkin et al addresses experimental study of surface acoustic waves (SAW) in a graphene-based inter-digital transducer (IDT). I believe that the general interest in this topic is in line with submission to Nanomaterials journal and potentially of broad interest.
However, the submitted manuscript in the present form should be improved. Consequently, I don’t recommend to publish this work in the present form.
The main weakness of the work is its lack of novelty. Numerous inter-digital transducer with graphene and with higher yields can be found in the literature (for instance references in this topical review https://doi.org/10.1088/1361-6463/aad593).
The authors have motivated their work “the use of graphene for the fabrication of acoustoelectronic devices is still difficult due to the high electrical resistance of graphene films compared to Al”. This is just plain wrong. Graphene has a much higher electronic conductivity than most other materials when suspended, or when encapsulated for example in h-BN (see for instance doi.org/10.1103/PhysRevLett.130.246201).
I recommend the authors to improve the graphene growth and transfer processes in order to achieve high quality graphene. Thus, in addition to encapsulate graphene within h-BN, an approach already demonstrated as highly critical in any electronic device fabricated with graphene.
In conclusion, considering all the comments above, I am inclined to reject the present work for publication in Nanomaterials in the present form.
Comments on the Quality of English LanguageTo be improved
Author Response
Dear Reviewer!
Thank you for your comments and helpful remarks. We have modified the text of the article accordingly.
Coment 1: The main weakness of the work is its lack of novelty. Numerous inter-digital transducer with graphene and with higher yields can be found in the literature (for instance references in this topical review https://doi.org/10.1088/1361-6463/aad593).
Response: In this paper, we are among the first to show the reduction of the loading on the crystal surface and the resulting increase in SAW velocity. The process of SAW visualization excited by MLG-IDT by scanning microscopy has been demonstrated for the first time. The optimal distribution of the acoustic field on the crystal surface without diffraction divergence is shown.
We know review https://doi.org/10.1088/1361-6463/aad593). This paper has information about the problem of electrical resistance of IDT electrodes. The possibility of doping graphene, which leads to an increase in electrical conductivity, has been considered. After improving the electrical characteristics, graphene returns to its original state in a short time.
Two approaches can be used to improve the electrical and respectively the acoustic characteristics of MLG-IDT. The first approach is related to the doping of graphene, which leads to an increase in the conductivity and a corresponding decrease in the resistivity of graphene. However, the doped state of graphene is metastable and cannot exist for a long time. Over time, graphene reverts back to its original state. Graphene encapsulation also fails to significantly improve the electrical properties of MLG-IDT. There is a second approach to improve the electrical and acoustic characteristics of MLG-IDT used in [19, 33], where the length of the MLG-IDT pins was reduced in order to reduce the IDT resistivity. In this case, the losses caused by high impedance are reduced, but there are large losses caused by the diffraction of the surface acoustic wave on the MLG-IDT aperture (the analog of light diffraction on a slit). In such a case, the acoustic wave will be divergent at the IDT output, resulting in high acoustic losses. Therefore, in the future, to create a graphene IDT, it is necessary to solve a serious problem on optimization of IDT parameters (aperture of IDT, number of electrode pairs, and resistivity of graphene).
Coment 2: The authors have motivated their work “the use of graphene for the fabrication of acoustoelectronic devices is still difficult due to the high electrical resistance of graphene films compared to Al”. This is just plain wrong. Graphene has a much higher electronic conductivity than most other materials when suspended, or when encapsulated for example in h-BN (see for instance doi.org/10.1103/PhysRevLett.130.246201).
Response: Here we disagree with remark. The resistivity of graphene in this paper is in agreement with values from other artickles. In this paper, the resistivity is ~4.2×10-6 Ω-m, while the values from other artickles are ~(4.0-5.0)×10-4 Ω-cm. It can be seen that these values coincide. The method of Raman spectroscopy was used at all technological stages. The spectra correspond to Fig. 1. No changes are observed. We tried using h-BN for encapsulation, but it does not significantly change the result, but it complicates the technology. As mentioned in the paper, the ideal is to find an optimal solution (graphene optimization, MLG-IDT structure optimization). Probably, the optimal solution is as in [10], when there is a correspondence between the crystal unit cell of the piezoelectric crystal and graphene. But this solution is only suitable for a group of materials of the langasite family, but not for other piezoelectric crystals. In this case, it is possible to grow graphene in a direct process on the crystal surface and not use the transfer.
Comment 3: I recommend the authors to improve the graphene growth and transfer processes in order to achieve high quality graphene. Thus, in addition to encapsulate graphene within h-BN, an approach already demonstrated as highly critical in any electronic device fabricated with graphene.
Response: The quality of graphene is consistent with that of graphene from other sources. This paper provides an impulse for research in this direction of acoustoelectronics. It is possible to search for other approaches when creating IDTs. The optimal variant is the one described in [10]. For us, the application of MLG-IDT structures in X-ray acousto-optics is of most interest [36]. The operation of the X-ray acousto-optical modulator is based on the application of electric fpulses to the IDT structure. MLG-IDT acts as a transparent diffraction grating with zero phase shift for X-ray radiation.

Round 2
Reviewer 2 Report
Comments and Suggestions for Authors
The revised version of the work entitled Application of graphene in acoustoelectronics by Roshchupkin et al has not made the necessary proposals for improvement to make it publishable. The changes made are merely cosmetic and do not substantially change the previous version.
In conclusion, considering all the comments above, I am inclined to reject the present work for publication in Nanomaterials.
Comments on the Quality of English LanguageNo comments
Author Response
Thank you for your correct remarks. Raman spectra of the original multilayer graphene and interdigital transducer are shown in Fig. 1. They are almost identical. Therefore, we note that the technological operations for fabrication of the MLG-IDT structure do not lead to changes in the properties of the multilayer graphene.